# Landsat-Derived Forel–Ule Index in the Three Gorges Reservoir over the Past Decade: Distribution, Trend, and Driver

**DOI:** 10.3390/s24237449

**Published:** 2024-11-22

**Authors:** Yao Wang, Lei Feng, Jingan Shao, Menglan Gan, Meiling Liu, Ling Wu, Botian Zhou

**Affiliations:** 1School of Geography and Tourism, Chongqing Normal University, Chongqing 401331, China; 2Chongqing Key Laboratory of Big Data and Intelligent Computing, Chongqing Institute of Green and Intelligent Technology, Chinese Academy of Sciences, Chongqing 400714, China; 3School of Information Engineering, China University of Geosciences, Beijing 100083, China

**Keywords:** Landsat 8 OLI, Forel–Ule index, driving mechanism, Google Earth Engine, Three Gorges Reservoir

## Abstract

Water color is an essential indicator of water quality assessment, and thus water color remote sensing has become a common method in large-scale water quality monitoring. The satellite-derived Forel–Ule index (FUI) can actually reflect the comprehensive water color characterization on a large scale; however, the spatial distribution and temporal trends in water color and their drivers remain prevalently elusive. Using the Google Earth Engine platform, this study conducts the Landsat-derived FUI to track the complicated water color dynamics in a large reservoir, i.e., the Three Gorges Reservoir (TGR), in China over the past decade. The results show that the distinct patterns of latitudinal FUI distribution are found in the four typical TGR tributaries on the yearly and monthly scales, and the causal relationship between heterogeneous FUI trends and natural/anthropogenic drivers on different temporal scales is highlighted. In addition, the coexistence of phytoplankton bloom and summer flood in the TGR tributaries has been revealed through the hybrid representation of greenish and yellowish schemes. This study is an important step forward in understanding the water quality change in a river–reservoir ecosystem affected by complex coupling drivers on a large spatiotemporal scale.

## 1. Introduction

Water quality deterioration has emerged as one of the most serious environmental problems in inland waters worldwide, posing severe threat to both public health and aquatic ecosystems [1,2,3]. Considering the complex impact of the natural and anthropogenic interaction processes in a watershed, water color can serve as a reliable indicator that intuitively reflects water quality deterioration [4,5,6]. Typically, the water color range is wide due to the apparent integration of chlorophyll, chromophoric dissolved organic matter, and suspended particulate matter, which are closely related to primary productivity, ecological health, and water contamination, respectively [7,8,9]. Specifically, higher chlorophyll concentration causes greener water, increased chromophoric dissolved organic matter leads to browning or blacking water, and more suspended particulate matter results in browner or yellower water [10,11,12]. Accordingly, water color can be preliminarily determined through the three optically active substances mentioned above [13,14,15]. However, as a result of the coupled complexity of three optically active substances and other factors (such as water depth), the unexplained and uncontrollable dynamics in water color can generally be difficult to quantify.

Conventional water color measurement relies on in situ chromaticity analysis, which cannot achieve the large-scale and high-frequency data acquisition [16,17,18]. Based on the early observation equipment (e.g., Forel–Ule colorimeter), the Forel–Ule coordinate which divides water color into 21 levels is constructed to achieve the conversion between remote sensing reflectance (R_rs_) spectrum and the Commission Internationale deléclairage (CIE) chromaticity system [19]. In recent years, the Forel–Ule index (FUI) with an interpolation/RGB model has been proposed using a Landsat 8 operational land imager (OLI) and Terra/Aqua moderate resolution imaging spectroradiometer, continuously proving that satellite multispectral imagery has provided a new alternative measurement for water color [20,21,22]. Meanwhile, FUI is widely utilized in satellite observation of water transparency, nutritional status, and phytoplankton bloom, demonstrating its broad application prospects in large lakes and vast oceans [23,24,25]. However, the distribution, trends, and drivers of the FUI shift in the complex scenarios remain prevalently elusive.

Due to the intermittent occurrence of phytoplankton bloom and summer flood in the Three Gorges Reservoir (TGR) in China, water color dynamics exhibits significant spatiotemporal heterogeneity, especially in the TGR tributaries with unique natural and anthropogenic interaction processes, such as abundant precipitation, strong sunshine, and dammed rivers [26,27,28]. Therefore, the TGR tributaries with the characteristics of a mixed river–reservoir system can serve as excellent field experimental sites to reveal the general laws of FUI shift [29].

By combining satellite and in situ observations, this study attempts to (1) characterize the spatial distribution and temporal trend of FUIs in the TGR tributaries using Landsat 8 OLI imagery batch processed on the Google Earth Engine (GEE) platform and (2) elucidate the natural and anthropogenic interaction processes that drive the identified FUI shift, particularly focusing on the quantitative analysis of temporal trends in the FUI, in order to confirm whether natural or anthropogenic drivers dominate the FUI trends in the TGR tributaries, alleviating the water quality deterioration.

## 2. Materials and Methods

### 2.1. Study Area

Since October 2010, the TGR has implemented a water level regulation strategy of storage (up to 175 m) in winter and discharge (down to 145 m) in summer [30], forming 38 primary tributaries with the drainage area exceeding 100 km^2^ and the tributary backwaters accompanied by different hydrodynamic conditions, e.g., deep–shallow lake, mainstream bay, lake–river transition, and river types in the Xiaojiang, Caotang, Daning, and Xiangxi Rivers, respectively (Figure 1) [31]. Differentiated hydrodynamic types may have led to strong heterogenous water color across different sections and seasons in the four typical TGR tributaries [32,33,34]. Moreover, the TGR, located in southwestern China, belongs to a subtropical monsoon climate with distinct four seasons, abundant cloud/mist, and low wind speed [35]. Under the influence of global warming, the TGR climatic characteristics shaped by year-round precipitation, abundant sunshine, and summer heatwave have become pronounced [36,37,38]. These have created a complex interaction between natural and anthropogenic processes in the four TGR tributaries.

### 2.2. Data Collection

The data utilized in this study consist of satellite and in situ observations. Landsat 8 OLI imagery, with nine high signal-to-noise ratio spectral wavebands and revisit cycle of 16 d [39], is selected as the satellite data to extract FUIs in the four TGR tributaries during the years of 2013–2023 (Figure 2). The surface reflectance (SR) derived from Landsat 8 OLI imagery has been extensively applied in the water color remote sensing [40]. The GEE platform provides free access to these data products, which have undergone radiometric correction and cloud masking using the Landsat SR code and c function of mask algorithm, respectively [41,42,43]. Additionally, the normalized difference vegetation index (NDVI) which reflects the intensity of natural processes (such as, mud–rock flow, summer heatwave, and lingering drought) and human activities (e.g., returning farmland to forest, urban expansion, and water level fluctuation) is derived from the daily products provided by the National Oceanic and Atmospheric Administration of the United States [44,45].

The in situ data encompass natural (including precipitation, sunshine duration, and air temperature) and anthropogenic (i.e., water level) drivers. Precipitation, sunshine duration, and air temperature are obtained from the National Meteorological Science Data Center. Water levels in the Xiaojiang, Caotang, Daning, and Xiangxi Rivers, controlled by the storage and discharge of the TGR, are sourced from the hydrological stations operated by the Hydrological Bureau of the Yangtze River Commission.

### 2.3. FUI Retrieval Method

The Landsat 8 OLI multispectral sensor utilized in this study captures only a portion of the electromagnetic radiation within a discrete spectral range; thus, the missing spectra must be reconstructed through linear interpolation. Consequently, the conversion equations for the triple stimulus values in the CIE-XYZ chromaticity system, which can substitute the three primary colors, are derived from the linearly weighted interpolation of a visible waveband of Landsat 8 OLI imagery, which is hereafter referred to as the interpolation model (Equation (1)) [46].
(1)$$\left\{\begin{array}{c} X=11.053R_{rs}\left(443\right)+6.950R_{rs}\left(482\right)+51.135R_{rs}\left(561\right)+34.457R_{rs}\left(655\right)\\ Y=1.320R_{rs}\left(443\right)+21.053R_{rs}\left(482\right)+66.023R_{rs}\left(561\right)+18.034R_{rs}\left(655\right)\\ Z=58.083R_{rs}\left(443\right)+34.931R_{rs}\left(482\right)+2.606R_{rs}\left(561\right)+0.016R_{rs}\left(655\right) \end{array}\right.$$

To characterize the 21 levels of FUI ranges in the visible waveband, the CIE formulates a two-dimensional chromaticity system, i.e., CIE-xy, where the horizontal and vertical coordinates are derived from the normalization of triple stimulus values [47]. The corresponding hue angle *α* is defined by rotating counterclockwise from the y-axis (Figure 3) and then estimated using Equation (2).
(2)$$\alpha =arctan2\left(x^{\prime },y^{\prime }\right)=arctan2\left(y-0.3333,x-0.3333\right)$$

The bias between multispectral and hyperspectral images in estimating the hue angle is addressed using the one-dimensional fifth-order equation (Equation (3)), which facilitates the FUI extraction in the TGR [48].
(3)$$∆\alpha =-52.16b^{5}+373.81b^{4}-981.83b^{3}+1134.19b^{2}-533.61$$
where b represents 1% of hue angle. The corrected hue angle is *α_t_ = α* + Δ*α*. Subsequently, the FUI can be categorized through the lookup table, according to Figure 3 [49].

### 2.4. Statistical Analysis for Driving Mechanism of FUI Trend

To evaluate the causal correlation between natural/anthropogenic drivers and te FUI, the Pearson correlation coefficient *r* and significance level *p* value are employed [21]. When setting the *r* and *p* thresholds, it is necessary to refer to similar TGR studies in order to select a moderate threshold, according to the multi-factor coupling of the FUI driving mechanism [22,23]. In this study, |*r*| ≥ 0.6 and |*r*| < 0.6 indicate that the responses of the FUI trend to precipitation, sunshine duration, air temperature, NDVI, and water level changes are significant and limited, respectively. The change rate of the FUI trend is quantified by the slope of the fitting line. The statistical significance of these relationships is determined by the *p*-level < 0.05. Furthermore, to quantify the relative contribution of natural and anthropogenic drivers to the FUI trend, a multiple general linear model is constructed [50]. In consideration of the huge gap between a complex real-world and simple theoretical scene, a single driver cannot play a particularly significant role in environmental change, and thus the relative contribution threshold should be lowered to evaluate the driver’s importance [33]. Therefore, if the relative contribution of a single driver is greater than 35%, then it is concluded to strongly regulate the FUI trend; if 10% ≤ contribution < 35% and both |*r*| ≥ 0.6 and *p* < 0.05 are satisfied, then this driver is also considered the dominant factor.

In order to achieve a clear attribution analysis for the spatial distribution and temporal trend of FUI, it can be classified into different color schemes to represent water color dynamics. According to the commonly used methods for large-scale water color remote sensing [51,52,53], a four-color scheme classification method, featuring blueish (1 ≤ FUI < 6), cyanish (6 ≤ FUI < 9), greenish (9 ≤ FUI < 13), and yellowish (FUI ≥ 13) schemes (Table 1), is relatively suitable for characterizing water color in the TGR tributaries and thus has been utilized.

## 3. Results

### 3.1. Spatial Distribution and Temporal Trend of FUI

Through the framework shown in Figure 4, latitudinal FUI distribution in the TGR tributaries can be extracted from the Landsat 8 OLI SR data batch processed on the GEE platform. Specifically, moderate latitudinal blocking, installed on the Xiaojiang, Daning, and Xiangxi Rivers with 30 pixels and Caotang River with 5 pixels, is applied for computational efficiency and result validity. Thus, the time series of latitudinal FUI distribution in the four typical TGR tributaries have been constructed by mean computation and chromaticity mapping (Figure 5).

Spatially, the FUIs in the four TGR tributaries exhibit significant instability, particularly along different latitudes, usually appearing at a low FUI upstream and high FUI downstream (Figure 5). Specifically, almost the whole inundation areas of the Xiaojiang and Xiangxi Rivers are in the greenish and cyanish schemes (moderate FUI), while the yellowish scheme mainly occurs near the estuary of the Caotang River (high FUI), and the blueish scheme (low FUI) primarily exists in the upstream of the Daning River. Overall, the time series of latitudinal FUI distributions in the Xiaojiang, Daning, and Xiangxi Rivers are relatively consistent, mostly belonging to the green and blue schemes (Figure 5a,c,d). In contrast, the Caotang River had a slightly higher FUI, with a greater proportion of yellow scheme, especially near the estuary (Figure 5b).

The yearly mean FUIs in the four TGR tributaries from 2013 to 2023 show high temporal heterogeneity (Figure 6). Specifically, the trend lines of yearly mean FUIs in the Xiaojiang, Daning, and Xiangxi Rivers point to 9–10, with the difference being that the change rates are 0.0015, 0.0466, and −0.0242, indicating that the three TGR tributaries are currently in the process of stabilizing, accelerating, and decelerating greening, respectively (Figure 6a,c,d); as a comparison, a trend line of yearly mean FUI in the Caotang River converges to 11 with a change rate of 0.1392, suggesting that the color scheme in this tributary is dramatically changing from greenish to yellowish (Figure 6b). It is worth noting that in 2020, the Xiaojiang, Caotang, and Daning Rivers present extremely low FUI values, while the Xiangxi River appears to have an extremely high FUI value; this negative asymmetry proves that the interannual FUI shifts in the four TGR tributaries are inconsistent. Moreover, the imagery lack caused by cloud cover may potentially bias the yearly mean FUI. Through sensitivity analysis, the slope of the fitting line slightly increases, and the outlier outside the confidence interval indistinctively decreases after excluding limited imagery years. The results indicate that cloud cover does not have a significant impact on the overall change rate of the FUI trend.

The monthly mean FUIs in the four typical TGR tributaries show a distinct seasonal pattern (Figure 7). From April to May, the FUIs increase is accompanied by an expansion in the greenish scheme and a decrease in the blueish scheme; the FUI peaks of 11.5–12.8 around June–July display an intermediate color between yellow and green; from August to October, the FUIs gradually decline and return to an alternating blueish–greenish state; from November to the following March, the FUI valleys of 6.9–9.8 occur with a blueish scheme dilation. Specifically, the Xiaojiang and Xiangxi Rivers show a single FUI peak in June; while the Caotang and Daning Rivers exhibit double FUI peaks in March and July/June, respectively. In addition, the Xiaojiang, Caotang and Daning Rivers display a double FUI valley feature, occurring in February/April and October/November, respectively; conversely, the Xiangxi River shows a single FUI valley in February. As shown in Figure 7a,b, the highest FUI peak and lowest FUI valley appear in the Caotang and Xiaojiang Rivers in July and February, respectively. Consequently, the seasonal FUI trends in the two tributaries upstream are more dramatic compared with the two tributaries downstream.

### 3.2. Natural and Anthropogenic Drivers of FUI Trend

The causal correlation between yearly mean FUI values and natural/anthropogenic drivers is presented in Figure 8a. The results indicate that the FUI is only related to water level, with *p* = 0.026 and *r* = −0.66, in the Xiaojiang River. Moreover, increases in precipitation, air temperature, or NDVI could potentially transform the FUI into an accelerated greening or even yellowing color on the yearly scale, due to positive *r* for all scenes, apart from precipitation in the Daning River. Overall, the yearly FUI trends in the four typical TGR tributaries display statistically diverse correlations with the natural and anthropogenic drivers.

Figure 8b exhibits the causal correlation between monthly mean FUIs and natural/anthropogenic drivers. Hence, advances in sunshine duration, air temperature, or NDVI could potentially transform the monthly FUI trends into an accelerated greening in the Xiaojiang, Daning, and Xiangxi Rivers or even yellowing in the Caotang River due to the strong positive relationship (*r* ≥ 0.6) and statistical significance (*p* < 0.05). It is worth noting that the water level appears to have the closest correlation with the FUI (*p* < 0.05 and |*r*| ≥ 0.6), indicating that the responses of monthly FUI trends in the four tributaries to the storage and discharge of the TGR are statistically significant.

Furthermore, a quantitative analysis of the relative contributions of natural and anthropogenic drivers to the synergistic effect on the temporal characteristics of FUI shift must be considered. Overall, the total contribution of precipitation, sunshine duration, air temperature, NDVI, and water level on the FUI trend ranges between 21.8% and 48.8% as well as between 68.8% and 97%, with a mean value of 36.2% and 88.8%, on the yearly and monthly scales, respectively (Figure 9), indicating the strong temporal heterogeneity in the driver contribution of FUI trend across the TGR.

Due to the high residual, lacking statistical significance, and weak correlation, no driver is found to play a dominant role in the FUI shifts in the four typical TGR tributaries on the yearly scale (Figure 9a). This result may be attributable to other drivers not included in this study controlling the yearly FUI trend.

On the monthly scale, air temperature plays an important role (contribution = 39.2%, 28.4%, 58.2%, 17.5%, *r* = 0.8, 0.62, 0.88, 0.89, and *p* = 0.0018, 0.016, 0.00018, 0.00011) in regulating the FUIs in the four TGR tributaries (Figure 9b). In contrast, water level had a slightly weaker performance (contribution = 33.3%, 13.5%, 52.2%, *r* = −0.94, −0.95, −0.91, and *p* = 0.0000045, 0.0000014, 0.00004), significantly impacting the FUI shifts in the Xiaojiang, Daning, and Xiangxi tributaries. Similarly, NDVI (contribution = 16.9%, 22.4%, 22.4%, *r* = 0.69, 0.65, 0.78, and *p* = 0.013, 0.023, 0.0028) greatly dominates the monthly FUI trend except for the Xiangxi River. Thus, we conclude that these natural and anthropogenic drivers have altered almost all the seasonal dynamics of water colors in the four TGR tributaries studied herein.

## 4. Discussion

### 4.1. Advantages and Limitations of Landsat-Derived FUI

Both RGB and interpolation models derived from the CIE-xy chromaticity system are commonly used approaches to retrieve FUI [54]. The RGB model achieves an objective representation of human color perception by superimposing the three primary colors. Due to computational issues arising from the negative values of the RGB model, the alternative interpolation model is defined to substitute the primary colors with the hypothetical colors. By integrating the visible R_rs_ spectra and color-matching functions, it is possible to estimate the triple stimulus values that serve as three primary colors for the human eye [55].

The RGB model that performed well within the FUI range of 1–7 relies solely on the red, green, and blue wavebands, limiting its ability to capture the water optical properties [56]. However, for the full range FUI calculation, the mean absolute error, root mean square error, and bias between the RGB and interpolation models are 3.46, 3.5, and −3, which are based on the random 1308 FUI measured samples; accordingly, there is a systematic underestimation in estimating the FUI with the RGB model. In contrast, the interpolation model compensates for missing wavebands, such as near infrared and shortwave infrared wavebands, more accurately reflecting the spectral complexity of eutrophic inland waters, e.g., the TGR tributaries.

In addition, the results of comparative analysis with 82 pairs of simultaneous field photography and true color imagery (sampling interval is less than 7 d and weather conditions are consistent) indicate that the RGB model tends to underestimate the FUI in favor of the blueish scheme, especially in river sections with high levels of eutrophication and algal density (Figure 10). Therefore, in studies examining the FUI shift across the TGR, the interpolation model can offer a reliable and accurate representation of water color dynamics.

### 4.2. Coexistence of Phytoplankton Bloom and Summer Flood

As shown in Figure 5, Figure 6 and Figure 7, the results indicate that latitudinal FUI distribution is not equal in the four typical TGR tributaries during 2013–2023. Specifically, blueish, cyanish, greenish, and yellowish schemes account for 5.7%, 37%, 44.8%, and 12.5% of the total inundation area, respectively. It is found that greenish (32.5%) and yellowish (16.4%) schemes concentrate in May–August and June–July, respectively, while the blueish (17.4%) and cyanish (43.7%) schemes tend to be in the months of winter and autumn, respectively. Furthermore, in terms of phytoplankton bloom and summer flood, the FUIs in the Xiaojiang and Caotang Rivers have the highest proportion of greenish (34.8%) and yellowish (18.3%) schemes, respectively.

During the summer of the past decade, the water colors in the four TGR tributaries have shown clear greening trends (for example, in the summers of 2013, 2016, 2018, and 2021, the FUIs are squeezed between 9 and 10, as shown in Figure 5), indicating the phytoplankton bloom occurrence in summer (consistent with the previous findings [33]), according to the corresponding color scheme of water quality characterization described in Table 1. Meanwhile, the water colors near the estuary of the Xiaojiang and Caotang Rivers tend to be yellowish (the FUIs are concentrated around 12–13), suggesting the emergence of summer flood containing a large amount of suspended sediment (coinciding with our previous findings [57]). Accordingly, the coexistence of phytoplankton bloom and summer flood is recorded by Landsat-derived FUI in the two TGR tributaries mentioned above, as shown in Figure 11.

The adaptation of phytoplankton bloom to summer flood caused by heavy rainfall could be attributed to the extreme summer heatwave, increasing terrestrial nutrient, and low flow velocity during the low water level running period [58,59]. Thus, further research on the anthropogenic controllable flow velocity can provide a scientific solution in the TGR ecological operation.

### 4.3. Implications for Further Studies

Our results show considerable impacts of phytoplankton bloom and summer flood on the FUI trend as indicated by the high probability of increased chlorophyll and suspended particulate matter concentrations. Chlorophyll concentration can commonly also indicate nutrient status. Because eutrophication can lead to a cascade of ecosystem crises, the increased chlorophyll concentration implies the high eutrophication risk in the TGR, which requires more future attention to the water quality deterioration caused by water storage. In addition, the high suspended particulate matter concentration induced by summer flood may affect the water quality by reducing the light penetration depth, increasing the pollutant solubility, and providing the toxic disinfection by-products. Thus, more future studies are needed to understand the underlying mechanisms about how the reservoir operation and local climate will affect the water quality.

As machine learning techniques play an important role in the automated feature extraction and nonlinear identification, the complex FUI trends in this study should represent an example of how natural/anthropogenic drivers alter water color dynamics. However, due to the limited study period of 2013–2023 and small amount of yearly and monthly mean FUIs, these parameters are not suitable for machine learning methods. Consequently, in the future, by increasing multi-source data and using machine learning techniques, our understanding of the driving mechanisms of water color dynamics could be deepened, and the accuracy and universality of FUI extraction could be improved.

This study mainly relies on multispectral imagery to extract a large-scale FUI shift, while hyperspectral data are applied as specific-site validation data. With the rapid development of satellite remote sensing, future studies can use sensors with high spatial, temporal, and spectral resolutions to distinguish the spectral responses of different water compositions [60], improving the precision and universality of the FUI model as well as the retrieval accuracy of water quality parameters under complex backgrounds, especially in the TGR tributaries.

## 5. Conclusions

The Landsat-derived FUI is applied to investigate the spatial distribution and temporal trend of FUIs in the four typical TGR tributaries from 2013 to 2023. Although there are large differences in the geomorphology, climate, hydrology, and other environmental conditions across the TGR, a general greening trend in latitudinal FUI distribution on the yearly scale is detected, except for the Caotang River, which is located close to the Yangtze River; the statistical analysis suggests that precipitation, sunshine duration, air temperature, NDVI, and water level are insufficient to explain the interannual FUI trend. In contrast, heterogeneity in the monthly FUI trend is related to natural and anthropogenic interaction processes, indicating that different drivers play various roles in terms of controlling the seasonal FUI shift. This study provides crucial insights to understand the water quality deterioration for geographers, limnologists, and reservoir managers.

## Figures and Tables

**Figure 1 sensors-24-07449-f001:**
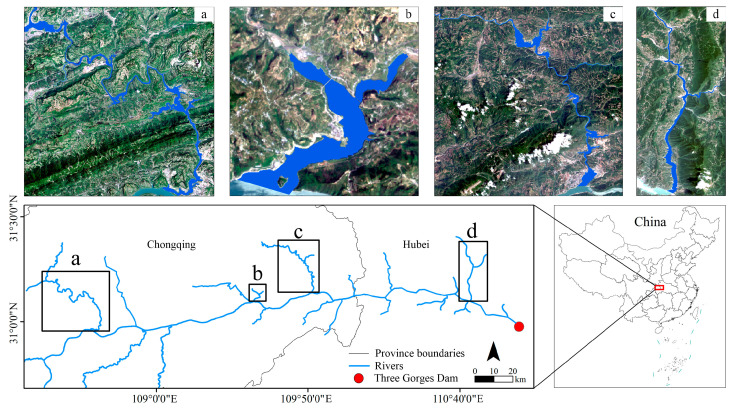
Locations of the four typical TGR tributaries, i.e., the (**a**) Xiaojiang, (**b**) Caotang, (**c**) Daning, and (**d**) Xiangxi Rivers.

**Figure 2 sensors-24-07449-f002:**
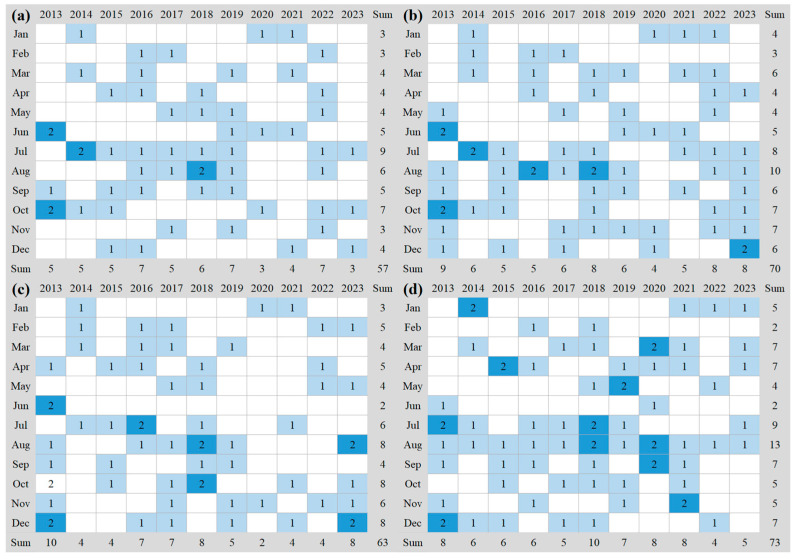
Sampling frequency of Landsat 8 OLI imagery covering the (**a**) Xiaojiang, (**b**) Caotang, (**c**) Daning, and (**d**) Xiangxi Rivers. The numbers 1 and 2 on the light and dark blue backgrounds respectively represent the number of satellite imagery available in the corresponding month. Blank indicates missing data.

**Figure 3 sensors-24-07449-f003:**
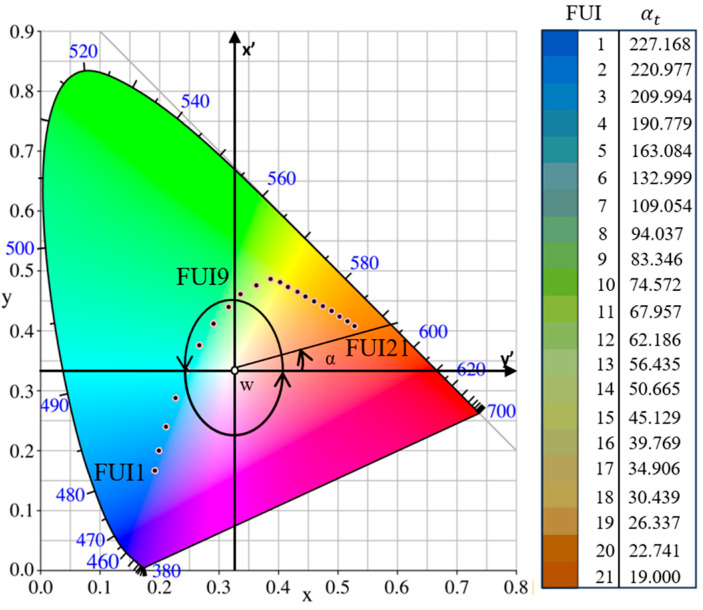
The 21 levels of FUIs and their hue angles α_t_ in CIE-xy chromaticity system.

**Figure 4 sensors-24-07449-f004:**
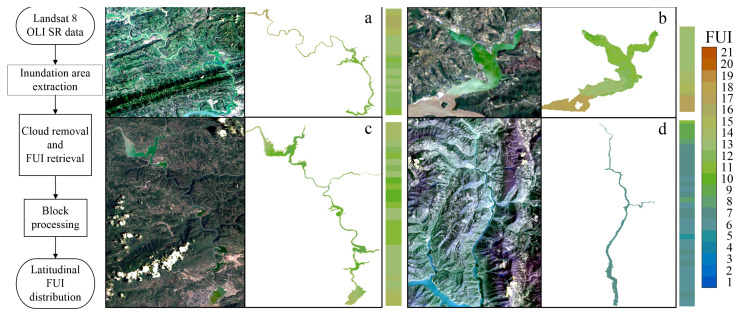
Framework of latitudinal FUI distributions in the (**a**) Xiaojiang, (**b**) Caotang (**c**) Daning, and (**d**) Xiangxi Rivers.

**Figure 5 sensors-24-07449-f005:**
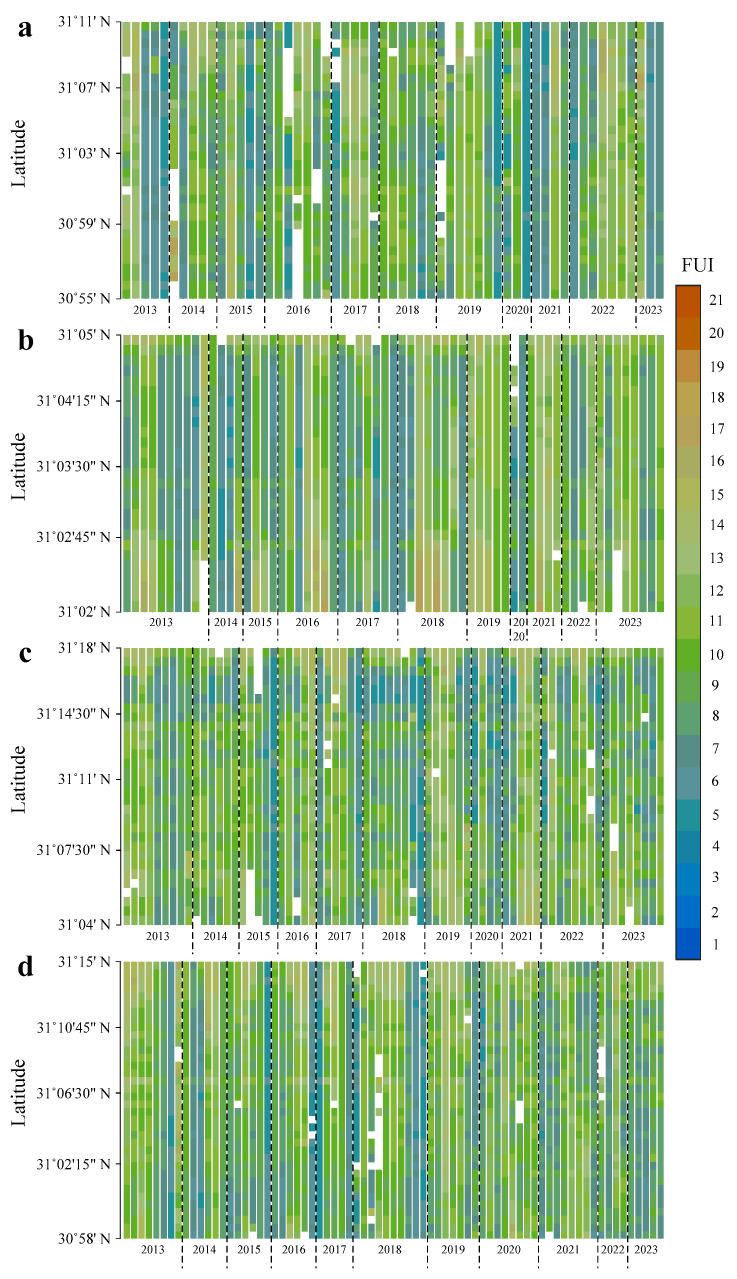
Time series of latitudinal FUI distributions in the (**a**) Xiaojiang, (**b**) Caotang, (**c**) Daning, and (**d**) Xiangxi Rivers. Blank indicates missing data due to cloud removal processing.

**Figure 6 sensors-24-07449-f006:**
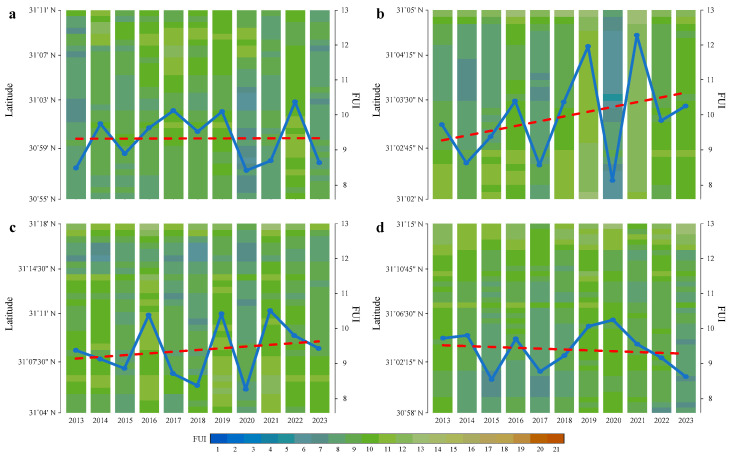
Latitudinal distribution of yearly mean FUIs in the (**a**) Xiaojiang, (**b**) Caotang, (**c**) Daning, and (**d**) Xiangxi Rivers. The blue line is connecting the yearly mean FUIs, and the red dashed line denotes the yearly FUI trend.

**Figure 7 sensors-24-07449-f007:**
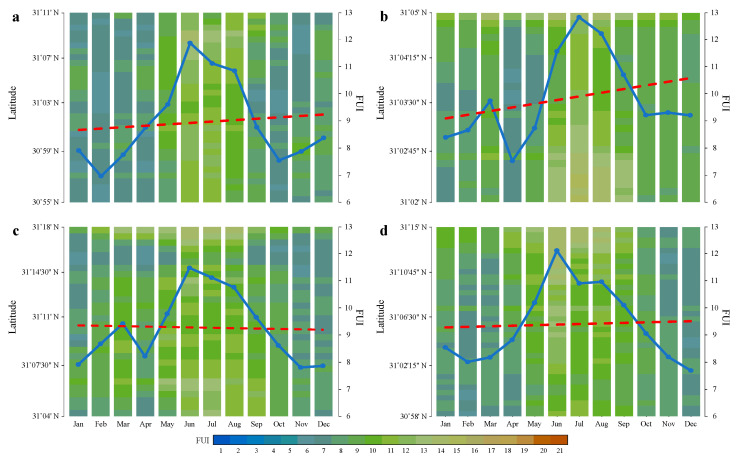
Latitudinal distribution of monthly mean FUIs in the (**a**) Xiaojiang, (**b**) Caotang, (**c**) Daning, and (**d**) Xiangxi Rivers. The blue line is connecting the monthly mean FUIs, and the red dashed line denotes the monthly FUI trend.

**Figure 8 sensors-24-07449-f008:**
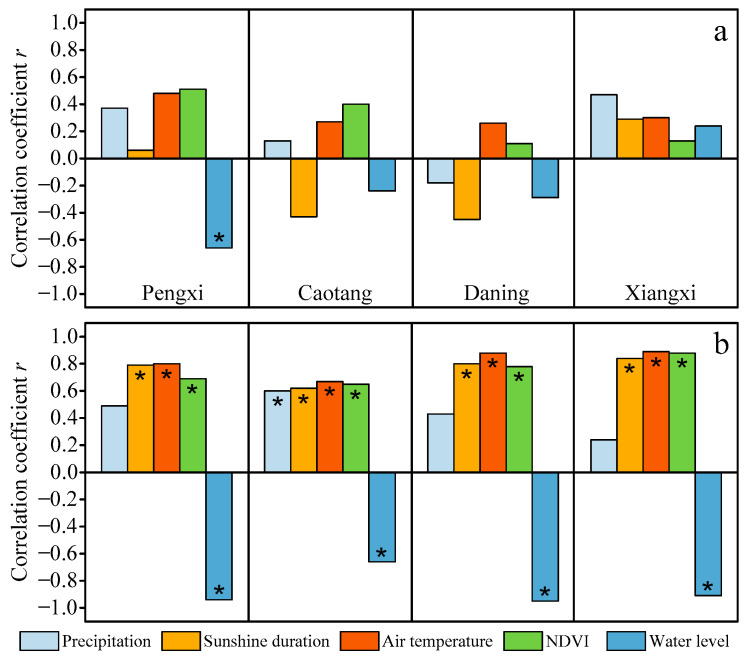
Causal correlations between natural/anthropogenic drivers and FUIs on the (**a**) yearly and (**b**) monthly scales in the four TGR tributaries. Asterisks denote the *p*-level < 0.05.

**Figure 9 sensors-24-07449-f009:**
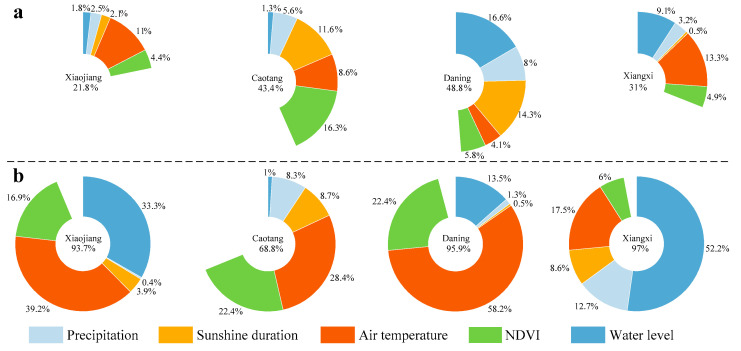
Contribution of natural and anthropogenic drivers to the FUI shifts on the (**a**) yearly and (**b**) monthly scales in the four typical TGR tributaries. Blank represents the residuals caused by drivers not considered in this study.

**Figure 10 sensors-24-07449-f010:**
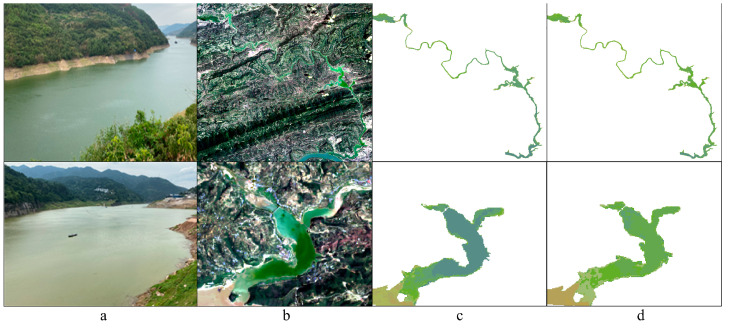
Comparative examples of (**a**) field photograph, (**b**) Landsat-derived true color imagery, Landsat-derived FUIs based on (**c**) RGB and (**d**) interpolation models for the water colors in the Xiaojiang (**upper**) and Caotang (**below**) Rivers on 18 May 2023.

**Figure 11 sensors-24-07449-f011:**
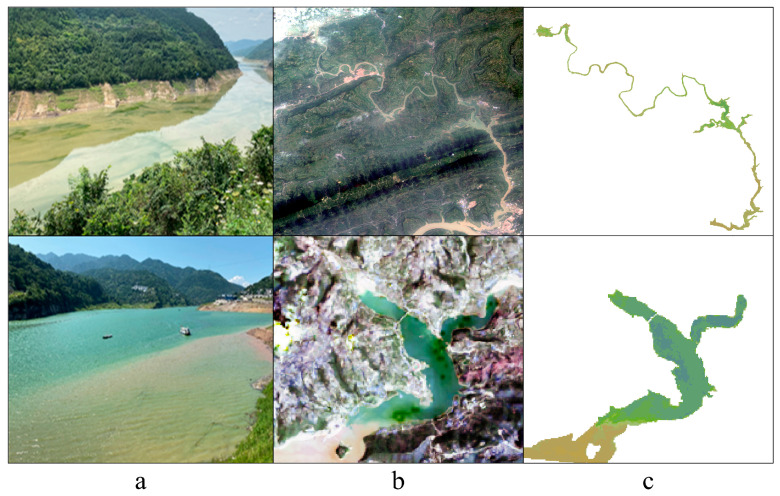
Examples of unique water color phenomenon, i.e., coexistence of phytoplankton bloom and summer flood, in the Xiaojiang (**upper**) and Caotang (**below**) Rivers based on (**a**) field photograph, (**b**) Landsat-derived true color imagery, and (**c**) Landsat-derived FUI on 4 June 2021.

**Table 1 sensors-24-07449-t001:** FUI range and corresponding water quality characterization.

FUI range	Color Scheme	Water Quality Characterization
1 ≤ FUI < 6	Blueish	Low levels of suspended sediment and algae density
6 ≤ FUI < 9	Cyanish	Moderate levels of dissolved organic matter and algae density
9 ≤ FUI < 13	Greenish	High level of algae density
FUI ≥ 13	Yellowish	High levels of suspended sediment and dissolved organic matter

## Data Availability

The data are available from the corresponding author on reasonable request.

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
