# Peer review of "Landsat-Derived Forel–Ule Index in the Three Gorges Reservoir over the Past Decade: Distribution, Trend, and Driver"

_sensors, 2024, doi:10.3390/s24237449_

Round 1
Reviewer 1 Report
Comments and Suggestions for Authors
Comments to authors:
1. Page 1, line 13: Water color is an essential indicator (rather than sentinel) of water quality assessment.
2. Page 1, line 17: “Google Earth Engine” should be written in capital letters.
3. Page 1, line 24-26: The last sentence of the abstract is largely a repetition of the sentence from lines 20-22.
4. Page 1, line 34: Same as above: Water color should be indicator, not sentinel of water quality deterioration.
5. Page 1, line 42-44: Other factors can also affect the water color, such as the depth of water body objects (rivers, reservoirs).
6. Page 2, line 67: “Google Earth Engine” should be written in capital letters.
7. Page 2, line 74-83: Add / cite more information about: Three Gorges Reservoir, its location, rivers regime, climate etc. How many tributaries does the Three Gorges Reservoir have? Were there any features that determined the selection of these four used in research?
8. Page 3, line 102-107: The Normalized Difference Vegetation Index is one of the most important remote sensing parameters, it was definitely not measured by in situ methods. Human impact has less of an impact on NDVI than natural processes.
9. Page 5-9: The description of the results is very good.
10. Page 11: Add line-space between lines 318 and 319.
Author Response
Reviewer #1:
Comment 1:
Page 1, line 13: Water color is an essential indicator (rather than sentinel) of water quality assessment.
Response:
Thanks for the suggestion. We have replaced “sentinel” with “indicator”.
Revision (Page 1, Line 11):
“Water color is an essential indicator of water quality assessment, …”
Comment 2:
Page 1, line 17: “Google Earth Engine” should be written in capital letters.
Response:
Thanks a lot for your careful reading and valuable reminder. We have replaced “google earth engine” with “Google Earth Engine”.
Revision (Page 1, Lines 15 and 25):
“Using the Google Earth Engine platform, …”
“… Google Earth Engine; …”
Comment 3:
Page 1, line 24-26: The last sentence of the abstract is largely a repetition of the sentence from lines 20-22.
Response:
Thanks for your reminding. Based on your suggestion, we have revised the last sentence of abstract so that the meanings expressed in the two sentences do not overlap with each other.
Revision (Page 1, Lines 22-24):
“This study is an important step forward in understanding the water quality change in river-reservoir ecosystem affected by complex coupling drivers on a large spatiotemporal scale.”
Comment 4:
Page 1, line 34: Same as above: Water color should be indicator, not sentinel of water quality deterioration.
Response:
Thanks for the suggestion. We have replaced “sentinel” with “indicator”.
Revision (Page 1, Line 32):
“…, water color can serve as a reliable indicator that intuitively reflects water quality deterioration [4-6].”
Comment 5:
Page 1, line 42-44: Other factors can also affect the water color, such as the depth of water body objects (rivers, reservoirs).
Response:
Thanks for your reminding. The influencing factors of water color are indeed not just the three elements (i.e., chlorophyll, chromophoric dissolved organic matter, and suspended particulate matter). I have made corresponding supplements in the revised manuscript, taking into account other factors, such as, water depth.
Revision (Page 1, Lines 41-42):
“… and other factors (such as water depth), ...”
Comment 6:
Page 2, line 67: “Google Earth Engine” should be written in capital letters.
Response:
Thanks a lot for your careful reading and valuable reminder. We have replaced “google earth engine” with “Google Earth Engine”.
Revision (Page 2, Line 66):
“… using Landsat 8 OLI imagery batch processed on the Google Earth Engine (GEE) platform, …”
Comment 7:
Page 2, line 74-83: Add / cite more information about: Three Gorges Reservoir, its location, rivers regime, climate etc. How many tributaries does the Three Gorges Reservoir have? Were there any features that determined the selection of these four used in research?
Response:
We really appreciate your valuable questions. We have added some relevant descriptions and references regarding the overview of the TGR to better showcase the situation of the study area for readers. In addition, we have explained that there are 38 TGR tributaries with a drainage area exceeding 100 km2 after the TGR is formed. The hydrodynamic conditions in these TGR tributaries present a variety of types, including deep-shallow lake, mainstream bay, lake-river transition, and river types, each showing different water color features. Therefore, this study selected the most representative tributaries from the four types of hydrodynamic conditions as the study objects.
Revision (Page 2, Lines 74-82):
“… [30], forming 38 primary tributaries with the drainage area exceeding 100 km2 and the tributary backwaters accompanied by different hydrodynamic conditions, e.g., deep-shallow lake, mainstream bay, lake-river transition, and river types in the Xiaojiang, Caotang, Daning, and Xiangxi Rivers, respectively (Figure 1) [31]. Differentiated hydrodynamic types may have led to strong heterogenous water color across different sections and seasons in the four typical TGR tributaries [32-34]. Moreover, the TGR, located in the southwestern China, belongs to a subtropical monsoon climate with distinct four seasons, abundant cloud/mist, and low wind speed [35].”
Newly added references:
- Xiao, Y.; Li, Z.; Guo, J.S.; Fang, F.; Smith, V.H. Succession of phytoplankton assemblages in response to large-scale reservoir operation: a case study in a tributary of the Three Gorges Reservoir, China. Monit. Assess. 2016, 188, 153.
- Huang, Z.L.; Li, Y.L. Water quality prediction and environmental capacity calculation in the Three Gorges Reservoir. China Water Resources and Hydropower Press 2006.
- Li, Z.; Wang, S.; Guo, J.S.; Fang, F.; Gao, X.; Long, M. Responses of phytoplankton diversity to physical disturbance under manual operation in a large reservoir, China. Hydrobiologia 2012, 684, 45-56.
Comment 8:
line 102-107: The Normalized Difference Vegetation Index is one of the most important remote sensing parameters, it was definitely not measured by in situ methods. Human impact has less of an impact on NDVI than natural processes.
Response:
Thanks a lot for your insightful comment to help us improve this work. We fully agree that Normalized Difference Vegetation Index is satellite remote sensing parameter and is not an anthropogenic driver. Thus, we have corrected the error that Normalized Difference Vegetation Index is in situ data and then remove it from the anthropogenic drivers.
Revision (Page 3, Lines 97-101; Page 4, Lines 107-108):
“Additionally, normalized difference vegetation index (NDVI) which reflects the intensity of natural processes (such as, mud-rock flow, summer heatwave, and lingering drought) and human activities (e.g., returning farmland to forest, urban expansion, and water level fluctuation) is derived from the daily products provided by the National Oceanic and Atmospheric Administration of the United States [44,45].”
“The in situ data encompasses natural (including precipitation, sunshine duration, and air temperature) and anthropogenic (i.e., water level) drivers.”
Comment 9:
Page 5-9: The description of the results is very good.
Response:
Thanks a lot for your recognition.
Comment 10:
Page 11: Add line-space between lines 318 and 319.
Response:
Thanks a lot for your careful reading and valuable reminder. We have added a line-space before “Author Contributions”.
Reviewer 2 Report
Comments and Suggestions for Authors
Please see the attachment for my comments and revise it accordingly.
Thanks

Author Response
Reviewer #2:
Paper titled “Landsat-derived Forel-Ule index in the Three Gorges Reservoir over the past decade: Distribution, trend, and driver”, here are some questions and suggestions for authors. Please read all the comments carefully and incorporate in revised version.
Question 1:
How does your research build on or differ from prior applications of the Forel-Ule Index in monitoring water quality in similar environments?
Response:
Thanks for the question. This study expands the FUI application, particularly focusing on the FUI applicability of complex river-reservoir system (i.e., the TGR). Unlike existing researches mainly focused on large lakes or/and open oceans, the study area of this study belongs to the inland waters affected by the coupling effects of multiple drivers, such as rainfall, sunshine, temperature, vegetation cover, and seasonal reservoir operation. Discussing the FUI applicability in complex river-reservoir system could provide scientific support for the reservoir ecological operation in the future.
Question 2:
Can you expand on the specific implications of your findings for reservoir management practices in the context of ecological conservation?
Response:
Thanks for the question. Our findings highlight the spatial distribution and temporal trend of FUI in the four typical TGR tributaries, which align closely with periods of phytoplankton bloom and summer flood. The results suggest that there is a strong correlation between the monthly mean FUI and the water level fluctuation caused by the storage and discharge of the TGR. Consequently, Landsat-derived FUI can be act as an indicator for phytoplankton bloom or summer flood and their periods when reservoir management should focus on mitigating nutrient inflow or accelerating turbidity removal. This study is an important step forward in understanding the water quality change in a large reservoir affected by water level management for ecological conservation.
Revision (Pages 11-12, Lines 329-339):
“Our results show considerable impacts of phytoplankton bloom and summer flood on the FUI trend as indicated by the high probability of increased chlorophyll and suspended particulate matter concentrations. Chlorophyll concentration can commonly also indicate nutrient status. Because eutrophication can lead to a cascade of ecosystem crises, the increased chlorophyll concentration implies the high eutrophication risk in the TGR, which requires more attention to the water quality deterioration caused by water storage. In addition, the high suspended particulate matter concentration induced by summer flood may affect the water quality by reducing the light penetration depth, increasing the pollutant solubility, and providing the toxic disinfection byproducts. Thus, more future studies are needed to understand the underlying mechanisms about how the reservoir operation and local climate will affect the water quality.”
Question 3:
Given the advancements in machine learning and AI for environmental monitoring, are there opportunities to integrate or compare these approaches with the FUI methodology?
Response:
Thanks for the question. We fully agree that the FUI retrieval model that integrates machine learning and AI will improve the technology level of water quality monitoring. For instance, machine learning algorithms could elevate the accuracy of FUI retrieval model by identifying patterns in water color dynamics influenced by complex non-linear drivers. In addition, comparing the FUI retrieval model with data-driven AI methods could help quantify predictive robustness of retrieval FUI and reveal complementary insights about water quality parameters across a large spatiotemporal scale. However, the experiment scope and data amount involved in this study are currently not suitable for using machine learning and AI. In future work, we will expand the experiment scope and data amount as well as focus on conducting studies in water color remote sensing based on machine learning and AI.
Revision (Page 12, Lines 340-347):
“As machine-learning techniques play an important role in the automated feature extraction and nonlinear identification, the complex response of FUI trend in this study should represent an example of how natural/anthropogenic driver alter water color dynamics. However, due to the limited study period of 2013-2023 and small amount of yearly and monthly mean FUIs, it is not suitable for machine learning methods. Consequently, in the future, by increasing multi-source data and using machine learning techniques, our understanding for driving mechanism of water color dynamics could be deepened and the accuracy and universality of FUI extraction could be improved.”
Question 4:
How do you see the FUI approach evolving with the introduction of higher-resolution or hyperspectral sensors, and how might these impact future studies on water quality dynamics?
Response:
Thanks for the question. We fully agree that the development of high spatial resolution of satellite sensors has improved the FUI applicability, enabling it to finely distinguish water color dynamics in small inland waters. High temporal resolution data can track the rapid water color dynamics, which is beneficial for exploring the driving mechanisms of FUI trends at different temporal scales. Furthermore, hyperspectral data can provide more detailed analysis of specific wavebands, enhancing the FUI sensitivity to subtle changes in water color parameters, such as, chlorophyll, chromophoric dissolved organic matter, and suspended particulate matter. Overall, satellite sensors with high spatial, temporal, and spectral resolutions will undoubtedly play a crucial role for monitoring water quality dynamics in the future.
Revision (Page 12, Lines 348-354):
“This study mainly relies on multispectral imagery to extract a large-scale FUI shift, while hyperspectral data is applied as specific-site validation data. With the rapid development of satellite remote sensing, future studies can use sensors with high spatial, temporal, and spectral resolutions to distinguish the spectral responses of different water compositions [60], thereby improving the precision and universality of FUI model, as well as the retrieval accuracy of water quality parameters under complex backgrounds, especially in the TGR tributaries.”
Suggestion 1:
Include a sensitivity analysis or mention potential biases introduced by missing data due to cloud cover in temporal trend assessments.
Response:
Thanks for the suggestion. For the temporal trend analysis of water color dynamics, missing imagery caused by cloud cover may potentially bias the yearly mean FUI. For example, in 2020 and 2023, the imagery number of the Xiaojiang River are less than 4, resulting in low mean FUIs in these two years, although the overall trend fitting uses mean calculation to alleviate the impact of missing data in individual years. To further investigate the impact of this bias, we conducted a sensitivity analysis and calculated the confidence interval for the yearly mean FUI, with a focus on displaying the FUI fluctuation range. The results showed that after excluding years with low image volume, the slope of FUI trend slightly increased, and the outlier number outside the confidence interval slightly decreased, indicating that the overall trends are not significantly affected.
Revision (Page 7, Lines 198-203):
“Moreover, the imagery lack caused by cloud cover may potentially bias the yearly mean FUI. Through sensitivity analysis, the slope of the fitting line slightly increases and the outlier outside the confidence interval indistinctively decreases after excluding limited imagery years. The results indicate that cloud cover does not have a significant impact on the overall change rate of FUI trend.”
Suggestion 2:
Consider adding a brief quantitative comparison of the RGB model versus the interpolation model for FUI estimation to substantiate the choice.
Response:
Thanks for the suggestion. For quantifying the accuracy differences of the interpolation and RGB models, we randomly collect 1308 pixels from Landsat 8 OLI imagery in the four TGR tributaries, extract their FUIs based on the two models, and then perform error analysis to understand the differences between the interpolation and RGB models at same sampling sites. The results showed that the mean absolute error, root mean square error, and bias are 3.46, 3.50, and −3, therefore there is a certain error between the RGB and interpolation models in retrieving the FUI value, and the RGB model systematically underestimates the FUIs. In addition, we conducted field investigations and then compared the retrieval results of two FUI models using 82 simultaneous field photographs. A significant blueish bias in the retrieval FUI using the RGB model is found. Therefore, in studies examining FUI shift across the TGR, the interpolation model can offer a reliable and accurate representation of water color dynamics.
Revision (Page 10, Lines 281-284 and 288-290):
“However, for the full range FUI calculation, the mean absolute error, root mean square error, and bias between the RGB and interpolation models are 3.46, 3.5, and −3, based on the random 1308 FUI measured samples; accordingly, there is a systematic underestimation in estimating FUI with the RGB model.”
“… 82 pairs of simultaneous field photograph and true color imagery (sampling interval is less than 7 d and weather conditions are consistent) …”
Suggestion 3:
Provide additional details on the threshold values used in the correlation and linear models, especially for distinguishing dominant drivers from non-significant factors.
Response:
Thanks for the suggestion. Considering the difference between the complex real world and simple theoretical scene in a complex river-reservoir system like the TGR, the driving mechanism of water color dynamics must be a nonlinear causal relationship model under the background of multiple coupling factors, such as meteorology, hydrology, and habitat. Therefore, when setting thresholds for r, p, and relative contribution in this study, we avoid using excessively high or/and low thresholds. Meanwhile, we referred to many similar studies and ultimately set the thresholds of r ≥ 0.6, p < 0.05, and relative contribution ≥ 35% to distinguish dominant drivers from non-significant factors.
Revision (Page 5, Lines 138-141, 147-150, and 154-157):
“When setting the r and p thresholds, it is necessary to refer to similar TGR studies in order to select a moderate threshold, according to the multi-factor coupling of the FUI driving mechanism [22,23].”
“In consideration of the huge gap between complex real world and simple theoretical scene, a single driver cannot play a particularly significant role in environmental change, and thus the relative contribution threshold should be lowered to evaluate the driver importance [33].”
“In order to achieve a clear attribution analysis for spatial distribution and temporal trend of FUI, it can be classified into different color schemes to represent water color dynamics. According to the commonly used methods for large-scale water color remote sensing [51-53], …”
Suggestion 4:
Expanding on the potential for integrating advanced machine learning models, particularly for driver identification, could increase the appeal of the study for a wider audience.
Response:
Thanks for the suggestion. We strongly agree that most advanced machine learning techniques will definitely help improve the study on the FUI driving mechanisms, especially for nonlinear driving mechanisms applicable to multi coupling factor in complex systems. Although this study has not yet used machine learning methods due to the small size of the data sample, we will definitely consider increasing the number and dimensions of the data sample in order to explore deeper potential driving mechanisms for water color dynamics using machine learning methods in the next step of research.
Revision (Page 12, Lines 340-347):
“As machine-learning techniques play an important role in the automated feature extraction and nonlinear identification, the complex response of FUI trend in this study should represent an example of how natural/anthropogenic driver alter water color dynamics. However, due to the limited study period of 2013-2023 and small amount of yearly and monthly mean FUIs, it is not suitable for machine learning methods. Consequently, in the future, by increasing multi-source data and using machine learning techniques, our understanding for driving mechanism of water color dynamics could be deepened and the accuracy and universality of FUI extraction could be improved.”
Suggestion 5:
Including a brief outlook on how hyperspectral data could further improve FUI analysis would provide valuable context and highlight future research opportunities.
Response:
Thanks for the suggestion. In current study, the FUI retrieval on a large spatiotemporal scale mainly relies on satellite multispectral imagery. In contrast, hyperspectral data is mostly used as specific-site validation data, making it difficult to achieve equivalent processing between the hyperspectral and multispectral imagery. However, obtaining finer information in water color waveband through hyperspectral resolution provides a more precise ability to identify the spatial distribution and temporal trend of FUI. Therefore, in the future, modeling using hyperspectral image data is necessary.
Revision (Page 12, Lines 348-354):
“This study mainly relies on multispectral imagery to extract a large-scale FUI shift, while hyperspectral data is applied as specific-site validation data. With the rapid development of satellite remote sensing, future studies can use sensors with high spatial, temporal, and spectral resolutions to distinguish the spectral responses of different water compositions [60], thereby improving the precision and universality of FUI model, as well as the retrieval accuracy of water quality parameters under complex backgrounds, especially in the TGR tributaries.”